# Synergistic Effect of Plasma-Activated Water with Micro/Nanobubbles, Ultraviolet Photolysis, and Ultrasonication on Enhanced *Escherichia coli* Inactivation in Chicken Meat

Kochakon Moonsub [1], Phisit Seesuriyachan [2], Dheerawan Boonyawan [3] and Wassanai Wattanutchariya [4,*]

1   Ph.D.'s Degree Program in Industrial Engineering, Department of Industrial Engineering, Faculty of Engineering, Chiang Mai University, Chiang Mai 50200, Thailand; kochakon_moo@cmu.ac.th
2   Division of Biotechnology, Faculty of Agro-Industry, Chiang Mai University, Chiang Mai 50100, Thailand; phisit.s@cmu.ac.th
3   Department of Physics and Materials Science, Faculty of Science, Chiang Mai University, Chiang Mai 50200, Thailand; dheerawan.b@cmu.ac.th
4   Advanced Technology and Innovation Management for Creative Economy Research Group, Department of Industrial Engineering, Faculty of Engineering, Chiang Mai University, Chiang Mai 50200, Thailand
*   Correspondence: wassanai@eng.cmu.ac.th; Tel.: +66-861177399

**Abstract:** The use of integrated plasma-activated water (PAW) with micro/nanobubbles (MNBs), ultraviolet (UV) photolysis, and ultrasonication (US) for the synergistic efficiency of *Escherichia coli* inactivation in chicken meat was investigated. A $2^k$ factorial design was employed to optimize the combined treatment parameters for pathogen disinfection in Design of Experiments (DOE) techniques. Its effectiveness was evaluated based on electrical conductivity (EC), oxidation–reduction potential (ORP), hydrogen peroxide ($H_2O_2$) concentration, and *E. coli* inactivation. The most significant impact on *E. coli* reduction was observed for MNBs, UV treatment time, and their interaction (MNBs and UV). Optimal *E. coli* inactivation (6 $\log_{10}$ CFU/mL reduction) was achieved by combining PAW with MNB and UV for 10 and 20 min, respectively. Integrating PAW with appropriate supplementary technologies enhanced *E. coli* inactivation by 97% compared to PAW alone. This novel approach provides a promising alternative for pathogen control in chicken meat, potentially improving food safety and shelf life in the poultry industry.

**Keywords:** plasma technology; plasma-activated water (PAW); microbubbles/nanobubbles; ultraviolet; ultrasonication; pathogen inactivation

## 1. Introduction

Poultry is a revenue-generating industry that continues to expand globally. Due to naturally existing microbes and enzymes, rotting food has a shorter shelf life and worse quality than fresh food. Traditional methods, like chemical disinfection and cold storage, while effective, often come with drawbacks such as residue buildup and a limited shelf life. This necessitates exploring innovative solutions to guarantee the quality and safety of poultry products while meeting the growing demand for fresh and diverse food options. Chemical disinfection also causes residues in the production process that affect the health of consumers. However, in order to maintain food quality, which affects consumer health, it is crucial to develop efficient antimicrobial treatments and apply them responsibly as the demand for meat grows. An increased consumption of food products that are linked to illness outbreaks has been promoted by the desire for a healthy lifestyle and customers' desires for more food diversity and accessibility, with a diet that is high in fresh foods, processed foods, and minimally processed foods. These goods are used without any processing or raw components that can introduce microorganisms [1]. Food-borne pathogens in the poultry industry, such as *S. aureus*, *C. perfringens*, *C. botulinum*, and *B. cereus*, can also enter the human food chain via contaminated poultry carcasses [2].

Plasma technology emerges as a promising alternative for microbial control in the poultry industry [3–8]. This technology utilizes cold ionized gas to generate reactive oxygen species (ROS), like ozone and hydroxyl radicals. These ROS possess potent antimicrobial properties, including *E. coli* [9,10], *S. aureus* [11–14], aerobic bacteria [15], *H. alvei, S. cerevisiae, L. mesentroids* [16], and *H. alvei* [17]. Therefore, a plasma gas release scheme has been developed to purify freshly washed poultry meat and prevent contamination. ROS, which are formed during the plasma process, cause damage to proteins, DNA, and enzymes that are essential for cell function. Additionally, ROS induce lipid peroxidation and amino acid oxidation in cell membranes. Therefore, it is crucial to efficiently eliminate microorganisms [18]. Fresh chicken meat is cleaned and sterilized using PAW, an application of plasma technology [19,20]. Recent studies have demonstrated the efficacy of plasma-activated water (PAW) in cleaning and sterilizing fresh chicken meat. However, the complex structure and thickness of poultry meat pose challenges to achieving uniform and complete disinfection. This necessitates further research and development to enhance PAW effectiveness, potentially through synergistic integration with other disinfection methods.

The related techniques found to have the potential to increase PAW efficiency include micro/nanobubbles (MNBs), ultraviolet (UV) photolysis, and ultrasonication (US). MNBs are tiny bubbles, ranging from micro- to nanometer-sized, that are filled with gas. These bubbles possess several crucial properties—including low buoyancy, a slow rising velocity, and a high interior gas density—that lead to prolonged suspension in water, maximizing their surface contact with pathogens [21–23]. MNBs with air can sustainably provide oxygen to the surrounding water due to there being much greater relative surface area and durability [24,25]. These characteristics are thought to aid in removing excited electrons and preventing the electron–hole pair from recombining [26]. Additionally, the single-scattering albedo of tiny bubbles in the solution is high and can enhance light scattering, potentially improving the efficacy of photocatalytic disinfection processes [27]. As a result, secondary radiation may be produced by electromagnetic illumination, which could cause electric charges close to the bubble to oscillate [28]. The wavelength will be longer as the bubble size decreases. MNBs may produce increased lateral/backward scattering intensity [28,29]. Essentially, the light efficiency of photocatalytic processes might be increased as a result of this light scattering effect. As far as we are aware, the impact of MNBs on photocatalytic disinfection and the underlying processes have not been researched. Due to their capacity to produce and retain reactive free radicals, MNBs have been used in food washing water treatment and waste water treatment in the food industry [30,31]. Secondly, UV is another type of disinfection method that has been demonstrated as an efficient method of eliminating pathogenic microorganisms [32,33]. The primary mechanism responsible for UV-induced photochemical reactions in microbial cells, specifically genetic components like DNA, is the inactivation of microorganisms [34–36]. Thirdly, the US technique utilizes high-frequency sound waves to generate cavitation bubbles that disrupt the cell membranes and internal structures of microorganisms. The US technique has been shown to be particularly effective in disinfecting thick and complex food structures. It has long been recognized for its effectiveness in washing food products because it can destroy germs through sonolysis and cavitation. US produces high pressure and a temperature gradient, causing air bubbles to form. These bubbles raise the fluid pressure outside, pushing liquid molecules into contact with one another. The mechanical shocks from these bubbles can disturb the structural and functional elements of cells to the point of cell lysis [37,38]. In the previous study, US significantly improved pathogen inactivation for food with thick and complex structures [19,20].

While PAW has emerged as a promising technology for food sanitation, currently, its efficiency in disinfection remains a bottleneck hindering widespread adoption in the food industry. This study proposes a novel approach to address this challenge by synergistically integrating PAW with MNB, UV, and US technologies.

Gas-filled bubbles enhance light scattering and oxygen supply using the technique of MNBs within the PAW reactor. Increased scattering improves light utilization, while

sustained oxygenation promotes the generation of ROS, which are crucial for pathogen inactivation. Additionally, MNBs potentially suppress electron–hole recombination, further boosting PAW efficiency. In the case of UV, radiation directly targets microbial DNA, causing irreversible damage and preventing replication from occurring. This synergistic action with PAW broadens the spectrum of targeted pathogens and ensures comprehensive disinfection. Lastly, ultrasound induces cavitation, generating shock waves and high shear forces that disrupt the cell membranes and internal structures of microorganisms. This mechanical action complements the oxidative effects of PAW and UV, and it is particularly effective against pathogens embedded within complex food matrices.

The advantages of the proposed combined approach were expected to be the synergistic action of PAW, MNB, UV, and US to significantly improve pathogen reduction compared to individual technologies. This could lead to shorter treatment times and lower energy consumption. The combination of technologies targets various microbial inactivation pathways—including oxidative damage, DNA disruption, and mechanical breakdown—expanding the range of treatable pathogens. This proposed approach might be potentially adaptable to various poultry processing stages and food types, offering wider applicability in the poultry industry. Additionally, optimizing the synergistic parameters of PAW, MNB, UV, and US for maximum efficacy, examining the underlying mechanisms of action and potential interactions between these technologies, and evaluating the economic feasibility and scalability of the combined approach for industrial implementation are areas that must be investigated.

This study presents a promising strategy to overcome the efficiency limitations of PAW disinfection in the poultry industry by integrating complementary technologies. The synergistic action of PAW, MNB, UV, and US holds the potential to revolutionize food sanitation practices, ensuring safer and healthier poultry products without the addition of external chemical disinfectants.

## 2. Materials and Methods

This experiment used the PAW system with a Flyback Transformer (FBT). In a previous study, this PAW system could produce an optimal $H_2O_2$ concentration with efficiently inactive food-borne pathogens [4]. The single-electrode non-thermal atmospheric pressure underwater plasma with a DC power supply was used to generate PAW in this experiment. The device consists of copper wire as a single electrode cover with a quartz tube and stainless steel as a ground. The reactor was connected to an FBT powered at 30 watts for 20 min at 25 °C. The device was set up to discharge beneath the water surface with an electrode–ground gap distance of 5 mm to create plasma discharge.

To investigate the effectiveness of the combined system on sterilization efficiency, PAW was used in conjunction with MNB, UV, and US. The operating conditions for the supplementary techniques were determined based on a previous study. The MNB system (made in-house) was set up at a pressure of 3 bar for 10 min at 25 °C [39]. In the meantime, the UVC lamp (OZUAR, Jiangsu, China) was installed and operated at 9 watts. Additionally, the system also utilized the US Digital Ultrasonic Cleaner Model 20A. The motor conductor had a length of 300 mm, and the ultrasonic power was 120 watts. The voltage was AC220-240V at a frequency of 50 Hz.

The operating temperature was 25 °C, and the full capacity was 3 L (OEM, Bangkok, Thailand). Figure 1a shows a schematic diagram of PAW combined with a supplementary technique, and Figure 1b shows a process diagram of PAW combined with a supplementary technique (b). These experimental conditions had a total of 22 experiments that were conducted based on a $2^k$ factorial design. Each experiment was repeated three times. Subsequently, the treated water was characterized in terms of EC, ORP, and $H_2O_2$ concentration. A test was conducted to determine the survival of *E. coli* after incubating for 48 h at 35 °C. The designed experiment was evaluated programmatically using the Analyze Factorial Design feature in Minitab® Statistical Software © 2021 Minitab, LLC, Pennsylvania State University, United States of America.

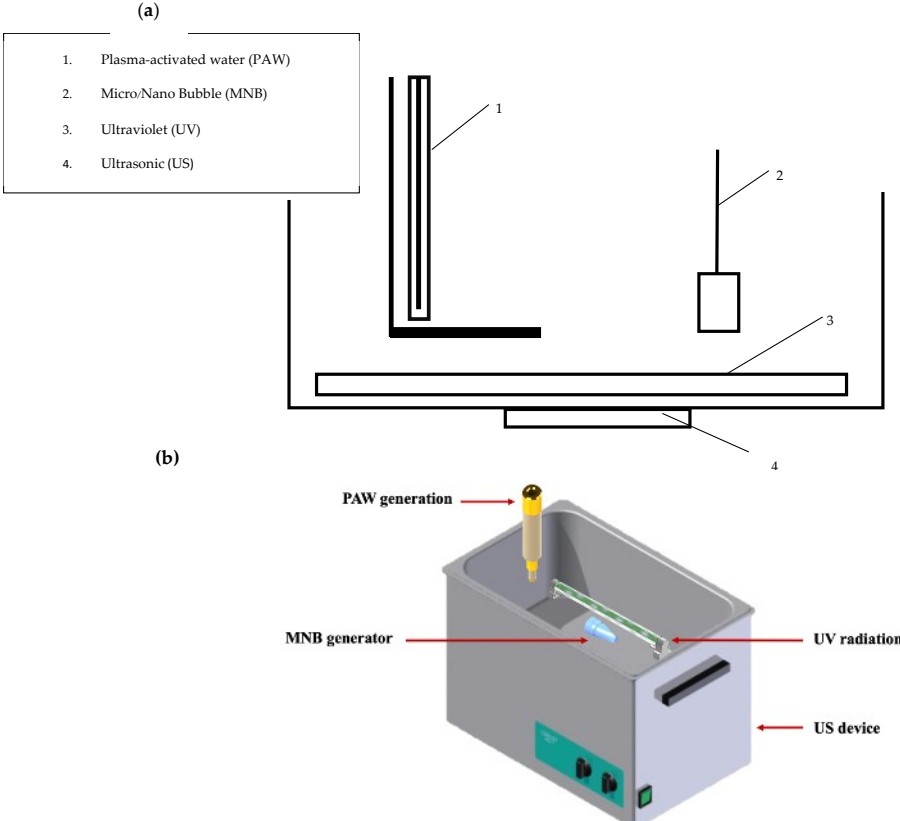

**Figure 1.** (**a**) A schematic diagram of PAW combined with a supplementary technique; (**b**) 3D model of PAW combined with a supplementary technique.

The $2^k$ factorial experiment was designed to determine the effect of plasma on enhancing efficiency when combined with other technologies. The experiment consisted of two parts: in the first part, a PAW generation experiment was conducted along with additional disinfection methods (MNBs and UV and US technology), and the second part used the soaking process, which involved using the optimum number of common food-borne pathogens. The optimal condition determined from the first experiment was then used to generate treated water, referred to as PAW'. In this second experiment, chicken meat contaminated with food-borne pathogens was tested during the soaking process.

Moreover, during the second experiment, PAW' was combined with other disinfection techniques (UV and US technology) to investigate the synergistic effect of these supplementary techniques on the inactivation of *E. coli*. The MNB technique was not performed on PAW due to the possibility of MNBs remaining in the water for an extended period during the soaking process, as shown in Figure 2. It was discovered that nanobubbles can last in aqueous solutions for several weeks after their creation. The solution contained bubbles with diameters ranging from 150 to 200 nm over the course of two weeks [40].

Theoretical knowledge and a review of related studies on important issues, such as common food-borne pathogens, PAW, MNB, and UV and US technology, are required to study the practical concerns for improving the effectiveness of plasma technology and supplemental disinfection procedures. To culture *E. coli* O157:H7, use a sterile technique to transfer a loopful of *E. coli* onto a slant in a culture flask containing 150 mL of Nutrient Broth (NB) as a starter. Incubate the flask at 37 °C with shaking at 150 rpm for 24 h. Then, using a sterile technique, pipette 10 mL of NB into another culture flask and incubate it at the same conditions for 24 h.

The concentration of *E. coli* cell cultures can be determined by measuring the spectrophotometer readings at $OD_{600} = 0.5$. An $OD_{600}$ reading of 1.0 corresponds to a concentration of $8 \times 10^8$ cells/mL in a 10 mL dilution sample, as shown in Figure 3a. In the soaking

procedure experiment, chicken meat was inoculated with a 10 mL dilution sample for 1 h, as depicted in Figure 3b.

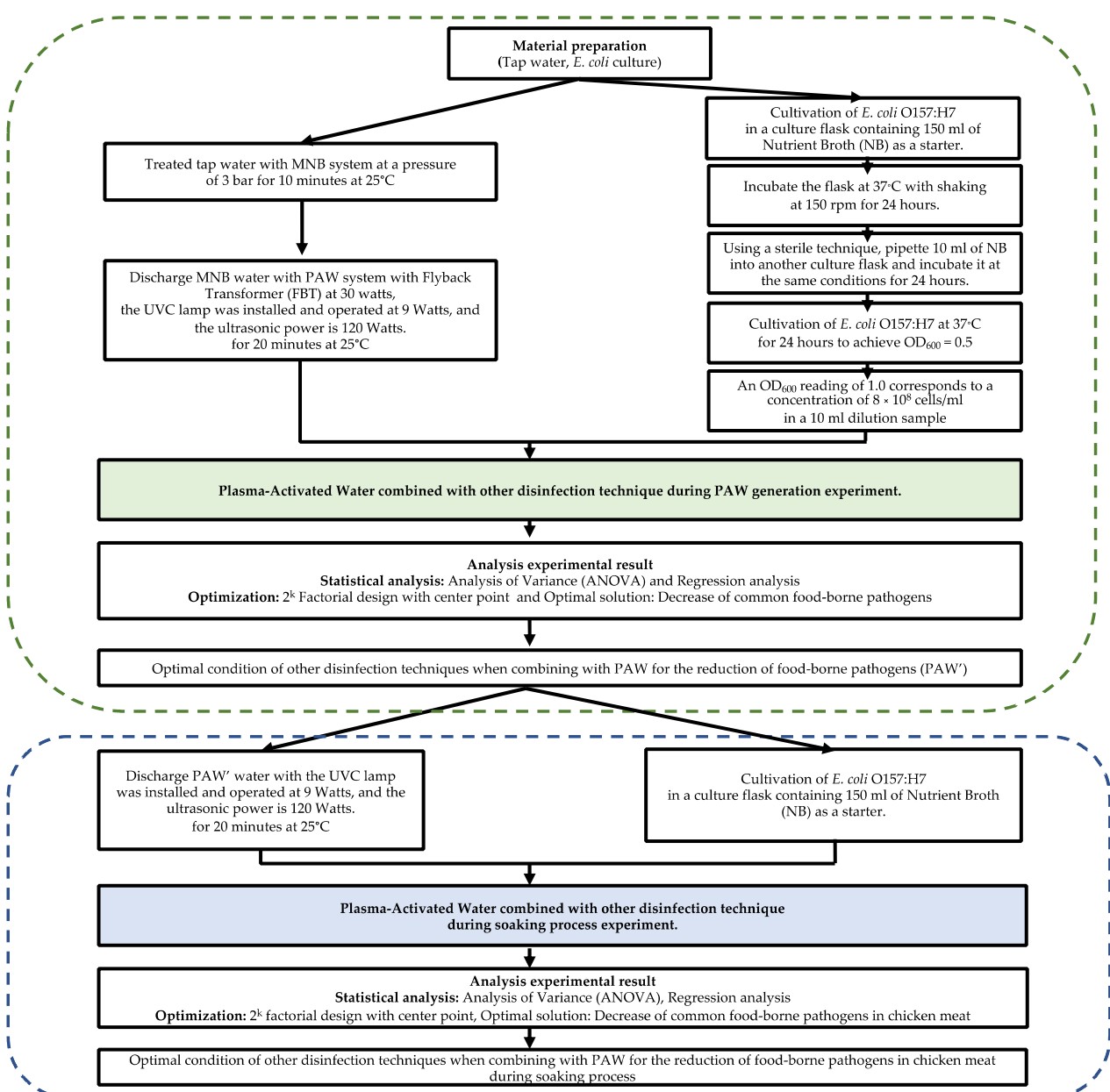

**Figure 2.** Scheme for PAW generation and soaking process experiment.

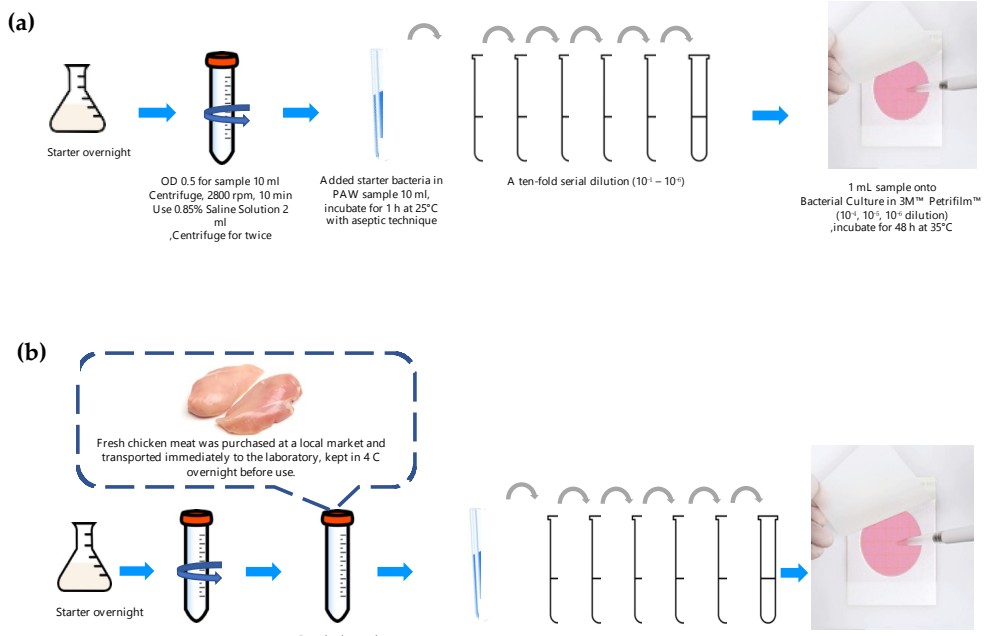

**Figure 3.** (**a**) An experimental schematic diagram of PAW generation process and (**b**) PAW with soaking process.

## 3. Results

To investigate the effect of PAW combined with supplementary techniques, the characterization of physicochemical properties and pathogen survival was analyzed. Experimental designs based on MNB, UV, and US combined with PAW were investigated using treatment time as a factor. There are high and low threshold levels for various factors. The results of the concept framework tests are presented below.

### 3.1. Effect of PAW Combined with a Supplementary Disinfection Technique during the Soaking Process on the Survival of E. coli in Inactivation Experiment

3.1.1. Physicochemical Characterization and Bactericidal Effects of PAW Combined with a Supplementary Technique

The physicochemical characterization and bactericidal effects of the PAW system combined with supplementary techniques are shown in Figure 4a–c. The results show that PAW with UV has the maximum concentration of EC, with a value of 240 mS/cm. Simultaneously, the ORP measurements show that PAW with MNB results in the highest ORP, which is 361.92 mV. Finally, the measurement of the $H_2O_2$ concentration indicates that PAW treatment with all combination techniques leads to the highest $H_2O_2$ concentration. The combination techniques in the treatment of PAW/MNB/UV and PAW/MNB/UV/US proved highly effective, resulting in a 6.00 $\log_{10}$ CFU/mL reduction in *E. coli*, as shown in Figure 4d. Furthermore, the inhibition of pathogenic microorganisms was optimized when using 10 min for the MNB treatment, 20 min for the UV treatment, and 20 min for the US treatment.

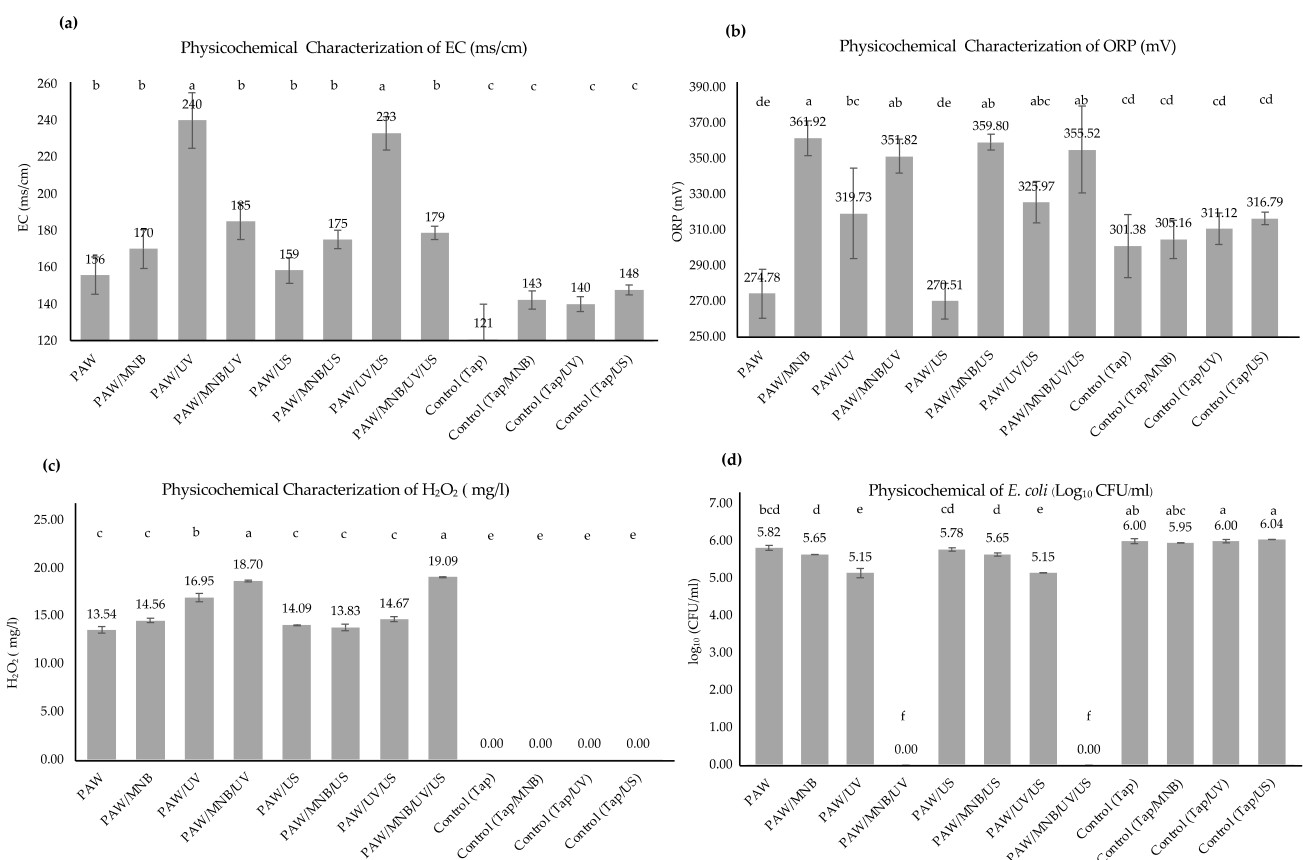

**Figure 4.** Physicochemical characterization of (**a**) EC, (**b**) ORP, and (**c**) $H_2O_2$ and (**d**) survival of *E. coli* in PAW combined with a supplementary technique during PAW generation. Survival of *E. coli* ($\log_{10}$ CFU/mL) after being treated with PAW combined with a supplementary technique. Error bars indicate standard deviation ($n = 3$). Different letters indicate significant difference among treatments ($p < 0.05$). Abbreviations: PAW = plasma-activated water; MNB = micro/nanobubbles; UV = ultraviolet photolysis; US = ultrasonication; Tap = tap water.

### 3.1.2. Statistical Evaluation of the Effect of PAW Combined with a Supplementary Technique on the Survival of *E. coli*

Table 1 presents the analysis of variance (ANOVA) for the factorial design experiment on PAW combined with a supplementary technique. The results indicate that only PAW combined with MNB, UV, and their two-way interactions (MNB*UV) significantly impact the survival rate of *E. coli* ($p < 0.05$). To determine the reliability of the model, the decision coefficient test (R-Square) was applied to the response values obtained from the ANOVA. The coefficients of determination for the *E. coli* quantification test, represented by $R^2$ and $R^2$ adjustment, were found to be 99.89% and 99.85%, respectively. As a result, the model is highly reliable and provides ample data to fit the equation and establish a predictive model capable of identifying the optimal type for reducing *E. coli* infection.

The coefficients of the elements that affect the reaction of the process are expressed in the form of an equation to create a predictive model. This predictive equation enables us to utilize the data obtained from the program's analysis in order to determine the coefficients of the terms that have a statistically significant impact. By employing Equation (1), we can identify the optimal values for the treatment factors that effectively minimize the survival of *E. coli*.

$$\text{Survival of } E.\ coli = 5.7975 - 0.01475\ \text{MNB} - 0.03237\ \text{UV} - 0.025013\ \text{MNB*UV} \quad (1)$$

According to the factorial design experiment, regression Equation (1) derived from this study can mathematically predict the responses for further study reproduction. The prediction

of responses is based on the change in significant factors and the coefficient of the equation. For instance, when the parameters of Equation (1) are set to lower conditions (MNB treatment time = 0 min, UV treatment time = 0 min), then the predicted survival of *E. coli* will be 5.79 $\log_{10}$ CFU/mL. On the other hand, if the experimental parameters are changed to upper conditions (MNB treatment time = 10 min, UV treatment time = 20 min), then the predicted survival of *E. coli* will be 0.00 $\log_{10}$ CFU/mL. These predicted values are insignificantly different from the experimental values of 5.82 and 0.00 $\log_{10}$ CFU/mL, respectively.

**Table 1.** ANOVA of *E. coli* inactivation experiment based on factorial experimental design.

| Source | DF | Adj SS | Adj MS | *F*-Value | *p* Value |
|---|---|---|---|---|---|
| Model | 4 | 92.7473 | 23.1868 | 2467.43 | 0.000 |
| Blocks | 1 | 0.0003 | 0.0003 | 0.03 | 0.860 |
| Linear | 2 | 67.7220 | 33.8610 | 3603.32 | 0.000 |
| MNB | 1 | 28.0635 | 28.0635 | 2986.38 | 0.000 |
| UV | 1 | 39.6585 | 39.6585 | 4220.27 | 0.000 |
| 2-Way Interactions | 1 | 25.0250 | 25.0250 | 2663.04 | 0.000 |
| MNB*UV | 1 | 25.0250 | 25.0250 | 2663.04 | 0.000 |
| Error | 11 | 0.1034 | 0.0094 | | |
| Total | 15 | 92.8507 | | | |

The results of the *E. coli* measurement tests were compared to determine the significant factor. The contour curve displays the MNB and UV treatment times, along with the optimal factor level value, to achieve the lowest possible amount of *E. coli*, as shown in Figure 4d. An *E. coli* inactivation test was conducted, and the results were used to create a contour plot illustrating the survival of *E. coli*. The contour plot between the MNB treatment time and UV treatment time in the plasma treatment, where the lowest result from the diagram represents the boundary of the desired area, is shown in Figure 5a. The boundary zone, depicted in dark blue, indicates a lower quantity of *E. coli*. After the plasma treatments with 20 min of MNB and then 20 min of UV, the quantity of *E. coli* was significantly reduced. Similarly, Figure 5b illustrates the effect of the MNB and US treatment times on the contour graph, while Figure 5c shows the effect of the UV and US treatment times. The lighter green boundary area indicates a lower survival population of *E. coli*. Furthermore, Figure 5b,c demonstrate that the US treatment time did not significantly affect the survival rate of *E. coli*, as all US treatment times projected the same *E. coli* population amount.

The response optimizer function was employed to determine the optimal conditions for the experiment. This function used a statistical approach to identify the appropriate parameters that contribute to the best response result. During evaluation, the program requires inputs for the direction, optimal value, weight, and importance. In this study, we set the minimum value of *E. coli* as the target value of 0, while the upper limit was defined as 6 based on the *E. coli* value in the control unit. Additionally, both the weight and importance were set to 1, as shown in Table 2. The results indicate that the most effective factors for reducing *E. coli* infection were an MNB treatment time of 10 min and a UV treatment time of 20 min, without US treatment. This combination resulted in a minimum *E. coli* value of 0.000 and a desirability rating of 1.0000. The high desirability highlights the significance of using the response surface analysis to determine the optimum point.

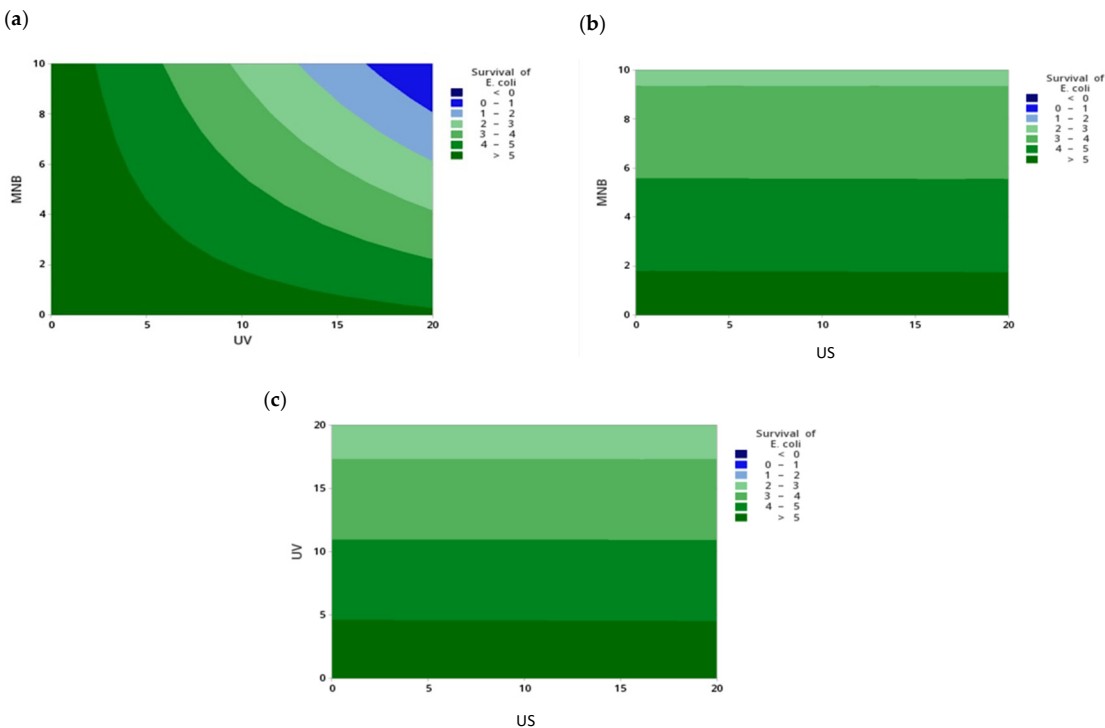

**Figure 5.** The contour plot on the survival of *E. coli* between (**a**) MNB treatment time and UV treatment time, (**b**) MNB treatment time and US treatment time, and (**c**) UV treatment time and US treatment time in PAW. Abbreviations: MNB = micro/nanobubbles, UV = ultraviolet photolysis.

**Table 2.** Optimal values of *E. coli* survival according to response optimization function.

| Parameters | | | | | | |
|---|---|---|---|---|---|---|
| **Response** | **Goal** | **Lower** | **Target** | **Upper** | **Weight** | **Importance** |
| Survival of *E. coli* | Minimum | | 0 | 6 | 1 | 1 |
| **Solution** | **MNB** | **UV** | **Survival of *E. coli* Fit** | | **Composite Desirability** | |
| 1 | 10 | 20 | −0.0000000 | | | 1 |
| **Multiple Response Prediction** | | | | | | |
| **Variable** | | | | | | **Setting** |
| MNB | | | | | | 10 |
| UV | | | | | | 20 |
| **Response** | **Fit** | **SE Fit** | **95% CI** | | | **95% PI** |
| Survival of *E. coli* | −0.0000 | 0.0485 | (−0.1067, 0.1067) | | | (−0.2385, 0.2385) |

### 3.2. Effect of PAW Combined with a Supplementary Disinfection Technique during the Soaking Process on the Survival of E. coli in Chicken Meat in Inactivation Experiment

3.2.1. Test Results for the Survival of *E. coli* in Chicken Meat

Figure 6 depicts the survival of *E. coli* in chicken meat when soaked with PAW from the first experiment, combined with supplementary techniques during the soaking process. The experiment utilized a factorial design technique ($2^k$ factorial design). UV and US treatments were implemented within a range of 0–20 min, as determined by previous investigations. The objective of this experiment was to assess the impact of PAW combined with supplementary techniques on the survival of *E. coli* in chicken meat. The survival rate of *E. coli* ($\log_{10}$ CFU/mL) was designated as the response variable, with the target value set to minimize during statistical evaluation. Each experiment was conducted three times, and the test results were characterized based on the survival of *E. coli*.

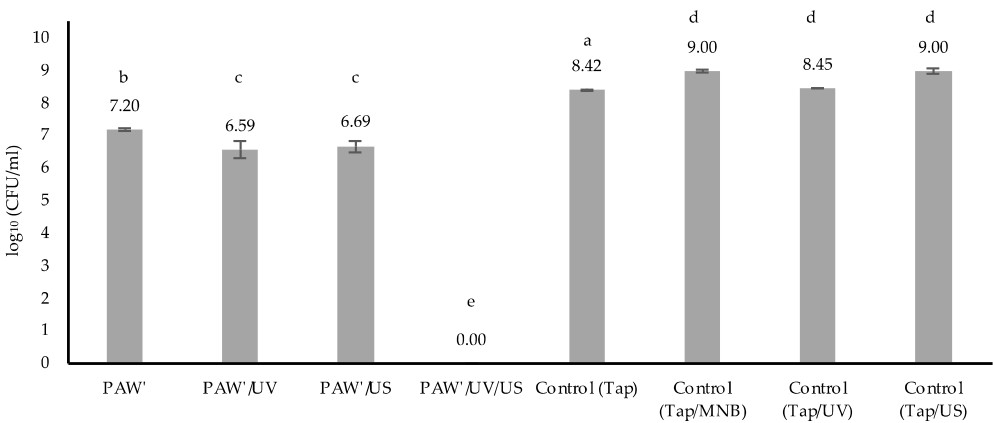

**Figure 6.** Survival of *E. coli* (log10 CFU/mL) in chicken meat experiment after being treated by PAW combined with supplementary techniques. Error bars indicate standard deviation (*n* = 3). Different letters indicate significant difference among treatments (*p* < 0.05). PAW′ = The optimal condition determined from the first experiment was used to generate treated water. Abbreviations: MNB = micro/nanobubbles; UV = ultraviolet photolysis; US = ultrasonication; Tap = tap water.

### 3.2.2. Effect of Activation Time on *E. coli* Survival in Chicken Meat

Table 3 shows the ANOVA results of the factorial design experiment on PAW combined with a supplementary technique. The results indicate that PAW combined with UV, US, and their two-way interactions (UV*US) significantly affect the survival rate of *E. coli* in chicken meat (*p* < 0.05). The reliability of the model was determined using the decision coefficient test (R-Square) of the response values obtained from the ANOVA. The coefficient of determination for the *E. coli* quantification test, denoted as $R^2$, and the adjusted $R^2$ value were 99.93% and 99.83%, respectively. Therefore, the model is highly reliable and provides sufficient data to fit the equation and create a predictive model that identifies the appropriate type for optimizing the reduction in *E. coli* infection in chicken meat.

**Table 3.** ANOVA of *E. coli* in chicken meat inactivation experiment based on factorial experimental design.

| Source | DF | Adj SS | Adj MS | *F*-Value | *p* Value |
|---|---|---|---|---|---|
| Model | 4 | 70.2937 | 17.5734 | 999.67 | 0.000 |
| Blocks | 1 | 0.0021 | 0.0021 | 0.12 | 0.752 |
| Linear | 2 | 51.7780 | 25.8890 | 1472.71 | 0.000 |
| UV | 1 | 26.6085 | 26.6085 | 1513.64 | 0.000 |
| US | 1 | 25.1695 | 25.1695 | 1431.78 | 0.000 |
| 2-Way Interactions | 1 | 18.5136 | 18.5136 | 1053.16 | 0.000 |
| UV*US | 1 | 18.5136 | 18.5136 | 1053.16 | 0.000 |
| Error | 3 | 0.0527 | 0.0176 | | |
| Total | 7 | 70.3465 | | | |

The coefficients of the variables that influence the response of the process are written as equations to construct a predictive model. The coefficients of the terms that had a statistically significant effect were determined from the equation values used to generate the prediction model, utilizing information from the program analysis. For instance, the following equation was employed to ascertain the most precise value and determine the minimum level of *E. coli* in chicken meat, which was examined and evaluated for the results. Equation (2) presents the formula for calculating the concentration of *E. coli* in chicken meat.

$$\text{Survival of } \textit{E. coli} = 7.1950 - 0.03025 \text{ UV} - 0.02525 \text{ US} - 0.015212 \text{ UV*US} \qquad (2)$$

According to the factorial design experiment, the regression Equation (2) derived from this study can mathematically predict the responses for further study reproduction.

The prediction of responses will be based on the changes in significant factors and coefficients of the equation. For instance, when the parameters of Equation (2) are set to the lower conditions (UV treatment time = 0 min, US treatment time = 0 min), the predicted survival of *E. coli* in chicken meat will be 7.19 $\log_{10}$ CFU/mL. On the other hand, if the experimental parameters are changed to the upper conditions (UV treatment time = 20 min, US treatment time = 20 min), then the predicted survival of *E. coli* in chicken meat will be 0.00 $\log_{10}$ CFU/mL. These predicted values are not significantly different from the experimental values of 7.20 and 0.00 $\log_{10}$ CFU/mL, respectively.

The response optimizer function was employed to determine the optimal conditions for the experiment. The data were analyzed and evaluated, and the most accurate number and the lowest level of *E. coli* were determined using Equation (2), which illustrates the formula for estimating the concentration of *E. coli* in chicken meat. From Table 4, the goal of the response was selected to minimize the value. The target value for *E. coli* is 0.00, while the upper limit is 8.11, which represents the mean of the *E. coli* test results in the control group without plasma treatment in this experiment. The results indicate that the most effective factors for reducing *E. coli* infection were a UV treatment time of 20 min and a US treatment time of 20 min. This led to a minimum *E. coli* concentration of 0.000 in chicken meat, with a desirability score of 1.0000. The high desirability score indicates that the optimum point analysis with the resulting surface was significant.

**Table 4.** Optimal values of *E. coli* survival in chicken meat according to response optimization function.

| Parameters | | | | | | |
|---|---|---|---|---|---|---|
| **Response** | **Goal** | **Lower** | **Target** | **Upper** | **Weight** | **Importance** |
| Survival of *E. coli* | Minimum | | 0 | 8.42 | 1 | 1 |
| **Solution** | **UV** | **US** | **Survival of *E. coli* Fit** | | **Composite Desirability** | |
| 1 | 20 | 20 | 0.0000000 | | 1 | |
| **Multiple Response Prediction** | | | | | | |
| **Variable** | | | | | | **Setting** |
| UV | | | | | | 20 |
| US | | | | | | 20 |
| **Response** | **Fit** | **SE Fit** | **95% CI** | | **95% PI** | |
| Survival of *E. coli* | 0.0000 | 0.0938 | (−0.2984, 0.2984) | | (−0.5168, 0.5168) | |

## 4. Discussion

This research investigated the potential of plasma technology to boost disinfection efficiency by comparing its effectiveness to established methods like MNB, UV, and US. In a two-phase study, researchers first experimented with PAW (plasma-activated water) to determine its effectiveness against common food-borne pathogens during a soaking phase. They then used the optimal PAW conditions identified in the first phase to treat water (labeled PAW') for the second phase. This treated water was used to soak chicken meat contaminated with a food-borne pathogen. Notably, MNB was not combined with PAW due to its prolonged persistence in water, potentially masking the effects of the other treatments. In the second phase, researchers also examined the combined effectiveness of PAW with UV and US technologies against a specific pathogen like *E. coli*, aiming to explore potential synergistic effects. To optimize this combined approach, they employed a factorial design, varying three treatment times: MNB, UV, and US. The research results from the experiments show that both the PAW/MNB/UV and PAW/MNB/UV/US conditions significantly reduced *E. coli* by 6.00 $\log_{10}$ CFU/mL. The most effective suppression of microbial pathogens occurred when using an MNB treatment time of 10 min, a UV treatment time of 20 min, and a US treatment time of 20 min. This study specifically focuses on ROS that are present in PAW, particularly long-lived ROS like $H_2O_2$, which are responsible for bacterial survival. The bacterium inactivation method can be explained using the following four steps.

Firstly, PAW is a unique disinfectant with significant antibacterial activity due to a variety of ROS. PAW results from non-thermal atmospheric plasma reacting with water and contains a wide range of highly ROS [41,42]. The underwater plasma generates a high concentration of ROS; secondly, ROS oxidize the lipid bilayer in the bacterial cell membrane, altering cell permeability and promoting membrane depolarization; in the third stage, ROS pass through temporary pores and induce oxidative stress within the cell, leading to an increase in intracellular ROS; and finally, in the fourth step, intracellular ROS interact with proteins, lipids, and carbohydrates, causing alterations in molecular structures and chemical bonds. However, it was discovered that adding organic matter to PAW reduces the levels of ROS. Excessive intracellular ROS, along with low pH, trigger redox reactions in the cell, disrupting pH homeostasis and resulting in cell death [11]. ROS play significant roles as signal molecules in different biological cells. It is known that ROS can boost inner oxidative stress and programmable cell death Furthermore, the research revealed synergistic antimicrobial properties within PAW [43]. The study found that PAW combined with specific supplemental approaches significantly affects its physicochemical properties and bactericidal potential. The physicochemical parameters of PAW were also studied, including EC, ORP, and $H_2O_2$. Synergistic effects between PAW and UV irradiation at 240 mS/cm were observed, as evidenced by the significantly elevated EC compared to other treatment conditions, which may be attributed to an increase in ionic activity. Conversely, PAW with MNB generated the highest ORP at 361.92 mV, indicating stronger oxidizing power. Interestingly, all PAW combinations resulted in higher $H_2O_2$ concentrations than PAW alone, potentially contributing to its improved antibacterial activity. As a result, PAW with combination techniques significantly reduced *E. coli* by 6.00 $\log_{10}$ CFU/mL. These findings correspond to previous research highlighting the effectiveness of PAW in bacteria inactivation. The results reveal that the PAW process can reduce *E. coli* by 0.74 $\log_{10}$ CFU/mL in chicken meat [20]. This consistency shows that the experimental results are moving in the same direction. Moreover, PAW with combinations can increase the efficiency of inhibiting germs more than what was achieved in past research.

Within this investigation, MNB technology was chosen due to its demonstrated efficacy in facilitating the transfer of reactive species generated by PAW. A subsequent analysis of the treatment process revealed that the combined application of PAW and MNB yielded the most substantial reduction in *E. coli* population, achieving a decrease of 0.35 $\log_{10}$ CFU/mL compared to the other tested conditions. Building upon this success, MNB was further synergistically combined with low-pH PAW to leverage the potential benefit of enhanced reactive species transfer alongside intensified bactericidal activity. To elucidate the specific factors contributing to this observed synergy, a comprehensive analysis of *E. coli* inactivation patterns under various treatment combinations was conducted. This analysis revealed that the synergistic effect primarily stemmed from the improved mass transfer dynamics of biochemically active species from the PAW matrix to the microbial targets within the solution, a finding that corroborates with those reported in previous studies [39,41,44,45]. MNBs, characterized by their high specific interfacial area, prolonged residence time, and elevated internal pressure, have demonstrably enhanced mass transfer rates from the gas phase to the liquid phase. Consequently, employing microbubbles to encapsulate the reactive species generated by cold atmospheric plasma processes (CAPP) presents a promising avenue for optimizing the efficiency of existing reactors. Within this study, planktonic bacterial cells were subjected to bactericidal PAW, which is abundant in ROS. These ROS possess the capability to interact with key microbial components—including membrane proteins, DNA, and metabolic enzymes—thereby disrupting vital cellular functions and structures and ultimately leading to potent antimicrobial efficacy.

Investigative research has demonstrably established, both empirically and theoretically, that Henry's law coefficient does not solely govern the dissolution of both highly and weakly soluble gases like $H_2O_2$ and $O_3$ [46]. ROS have been identified as the primary inactivation agents in non-thermal plasma processes. Previous research has aptly highlighted the crucial role of ROS within PAW, particularly highlighting the impact of exceptionally

long-lived ROS, such as $H_2O_2$, on bacterial survival. The synergistic effect of a low pH and excessive intracellular ROS accumulation triggers redox reactions within the microbial cell, disrupting pH homeostasis and ultimately leading to cell death [11,47–50]. The primary inactivation agents are ROS. The PAW production process generates a significant quantity of ROS. The production of ROS affects the physicochemical properties of PAW with MNBs. The following describes the series of key reactions ultimately leading to the formation of $H_2O_2$ [51]. The rise in ORP was most likely caused by PAW, as MNBs included more active ions and oxidizing species [52]. The relationship between pH and ORP is noteworthy. In acidic environments, increased hydrogen ion ($H_+$) concentration directly influences ORP.

The present study investigated the effect of storage time on the physicochemical properties of plasma-activated water (PAW), particularly focusing on its ORP and $H_2O_2$ content. This research observed that the ORP values increased over a 2 h storage period, suggesting an ongoing production of oxidizing species within the PAW matrix. To elucidate the underlying mechanisms, the $H_2O_2$ content was measured as a key indicator of ROS generation. As outlined in Equations (3)–(5), $H_2O_2$ is primarily produced through water dissociation and subsequent OH radical recombination. However, reactions 1 and 2 also contribute to $H_2O_2$ consumption over time, leading to a gradual decrease in its concentration during storage [52].

$$H_2O_2 \rightarrow HO\bullet + HO_2\bullet + O_2 \tag{3}$$

$$NO^- + HO \rightarrow NO^- + HO \tag{4}$$

$$HO\bullet + HO\bullet \rightarrow H_2O_2 \tag{5}$$

Both the PAW/UV and PAW/UV/US treatments decreased *E. coli* by 0.85 $\log_{10}$ CFU/mL in the generated process. As a result, in the UV and US soaking conditions, *E. coli* had a survival population of 1.83 $\log_{10}$ CFU/mL in chicken meat. Due to the major mechanism of bacterial inactivation by UV radiations, the dimerization of thymine bases occurs in their DNA strands, which has been discussed in relation to how UV radiation affects the formation of PAW [53]. The UV radiation produced by the direct discharge of plasma underwater helped to inactivate germs [54]. The intensity of UV radiation generated in the underwater discharged plasma increases with PAW conductivity [53,55].

Another technique used to decontaminate spores resembling the *Bacillus subtilis* strain MW01 is ultraviolet (UV) irradiation [56,57]. Several strategies have been explored to enhance ROS production within MNBs, including the introduction of UV radiation or chemical additives like copper [58–60]. However, the underlying mechanisms of this synergy and its impact on microbial inactivation remain unclear. This study aimed to investigate the combined effect of PAW and UV irradiation on *E. coli* survival compared to individual PAW and UV treatments.

Notably, synergistic antimicrobial activity has been previously observed in PAW combinations [43]. For instance, a study reported a >5 $\log_{10}$ CFU/mL reduction in *E. coli O157:H7* populations following exposure to UVA light for 30 min [61]. Nonetheless, the effects of combined PAW and UV treatments on the microbial load and quality parameters of fresh produce have not yet been fully elucidated. Possible mechanisms for microbial inactivation in this system involve both direct and indirect pathways. Direct inactivation can occur through UV-induced DNA damage, as UV light with a wavelength of approximately 260 nm can induce thymine–cytosine dimer formation, thus hindering DNA replication [62,63].

Additionally, the electric field ions and UV radiation generated by plasma can influence the biological activity of the resulting excited molecules and free radicals. Indirect inactivation, typically mediated by ROS formed in water, is believed to contribute significantly to radiation-induced cell death, as demonstrated by clonogenic survival experiments [64]. Equations (6)–(15) further illustrate the complex interplay between ionizing radiation, free radical formation, and subsequent ROS generation. While studies have explored the use of UV radiation and chemicals to boost ROS production in MNBs, the

detailed mechanisms underlying this synergy and its practical implications for microbial inactivation—particularly in food applications—warrant further investigation.

This study aims to address this knowledge gap by examining the combined effect of PAW and UV treatments on *E. coli* survival and its potential for enhancing the disinfection efficacy in chicken meat, which currently faces limitations due to its thickness, complex structure, and fat content. The efficacy of plasma-activated water (PAW) as a disinfectant can be further enhanced by introducing additional factors that promote the generation of ROS. These include UV radiation and chemical catalysts, like copper, as demonstrated in previous studies [58–60]. The synergistic effect of such interventions has been quantified through controlled comparisons of *E. coli* survivability following PAW-UV treatment versus a standard inactivation test. This approach allows for the construction of predictive models that estimate the impact of combined therapies on bacterial populations. Supporting this strategy is the established synergistic antimicrobial activity of PAW, as identified in independent research [43].

Additionally, UV light—especially UVA within the 260 nm wavelength range—exhibits significant bactericidal properties. In one study, *E. coli* O157:H7 populations were reduced by over 5 $\log_{10}$ CFU/mL after exposure to UVA light for 30 min [61]. However, the combined application of PAW and UV to address the microbial burden and quality parameters in fresh produce remains unexplored. One proposed mechanism for microbial inactivation through PAW-UV treatment involves direct DNA damage inflicted by UV rays emitted by the plasma [62]. This phenomenon, which is well documented in UV radiation studies, disrupts DNA replication by inducing thymine and cytosine dimerization within the same DNA strand [63].

UV Radiation Excitation

$$H_2O \rightarrow H_2O^* \rightarrow {}^\bullet OH + H^\bullet \tag{6}$$

UV Radiation Ionization

$$H_2O \rightarrow H_2O^\bullet + e^- \tag{7}$$

$$H_2O + H_2O^\bullet \rightarrow H_3O^+{}_{aq} + {}^\bullet OH \tag{8}$$

$$e^- + H_2O \rightarrow OH^- + H^\bullet \tag{9}$$

$$e^- + nH_2O \rightarrow e^-{}_{aq} \text{ (aqueous electron)} \tag{10}$$

UV Radiation Recombination of Ion Radiations

$$H_3O^+{}_{aq}\ e^-{}_{aq} \rightarrow H^\bullet + H_2O \tag{11}$$

$${}^\bullet OH + OH^\bullet \rightarrow H_2O_2 \tag{12}$$

$$H^\bullet + H^\bullet \rightarrow H_2 \tag{13}$$

UV Radiation in the Presence of Oxygen

$$O_2 + e^-{}_{aq} \rightarrow O_2^{\bullet -} \tag{14}$$

$$O_2 + H^\bullet \rightarrow HO_2^\bullet \leftrightarrow H^+ + O_2^{\bullet -} \tag{15}$$

Ultrasound

$$H_2O \rightarrow {}^\bullet OH + H^\bullet \tag{16}$$

$${}^\bullet OH + {}^\bullet OH \rightarrow H_2O_2 \tag{17}$$

$$H^\bullet + H^\bullet \rightarrow H_2 \tag{18}$$

$$OH + {}^\bullet OH \rightarrow {}^\bullet O^\bullet + H_2O \tag{19}$$

$${}^\bullet O^\bullet + N_2 \rightarrow {}^\bullet NO + {}^\bullet N \tag{20}$$

$$O_2 + H^\bullet \rightarrow HO_2{}^{\bullet -} \leftrightarrow H^+ + O_2{}^{\bullet -} \tag{21}$$

Despite the promising results of PAW-UV treatment on *E. coli* in chicken meat, further research is necessary to optimize its efficacy for practical applications. US technology has been explored as a potential enhancer, with the rationale that high temperatures and pressures generated by cavitation bubbles could promote the thermal dissociation of water and enhance ROS production. However, studies have shown that ROS production by US is dependent on exceeding the cavitation threshold (Equations (16)–(21)) [52]. In our study, while the combined PAW-US treatment demonstrated a slight decrease in *E. coli* population ($0.22 \log_{10}$ CFU/mL) compared to PAW alone, it did not significantly outperform either method. This suggests that further investigation is needed to optimize the parameters of US application for synergistic effects with PAW.

The observed differences in bacterial survival between chicken muscle and skin could be attributed to variations in their composition. The higher organic matter content on the skin likely hindered the penetration of ROS produced by PAW-US, resulting in less severe membrane damage and bacterial protrusion compared to the muscle. Furthermore, lipid oxidation, a known concern associated with plasma treatments in muscle foods [20], may have been exacerbated by the US-induced disruption of bacterial membranes, releasing unsaturated lipids prone to free radical interactions. Therefore, future studies should consider mitigation strategies for lipid oxidation to maintain the quality of chicken meat treated with PAW-US.

Our findings align with those of previous reports highlighting the synergistic potential of combining PAW with other non-thermal technologies, such as moderate heat and ultrasound, for enhanced antibacterial efficacy [20,65]. This approach holds promise for reducing food-borne pathogens while minimizing negative impacts on product quality. For instance, the sequential application of PAW and moderate heating (60 °C) effectively reduced injected pathogens on shredded cabbage without compromising its quality [65]. Similarly, recent research has demonstrated the effectiveness of PAW-US for inactivating *E. coli* on chicken meat and skin [66,67].

PAW combined with other techniques is larger than PAW alone because the major mechanism of bacterial inactivation can be enhanced by the synergy of these technologies, such as the following: PAW with MNB. Using a combination of techniques can improve the efficiency of biochemically reactive species mass transfer from plasma to microbial targets in a solution [44]. UV radiations lead to the dimerization of thymine bases in their DNA strands, which has been discussed in relation to how UV radiation affects the formation of PAW [68]. The UV radiation produced by the direct discharge of plasma underwater helped to inactivate germs [54]. UV radiation emitted by the direct discharge of plasma underwater aided in the inactivation of microorganisms.

Moreover, US helps accelerate the rate of PAW penetration into the samples. Accelerated oxidation processes (AOPs) based on US and UV radiation are also receiving scientific attention for water treatment and disinfection. When UV and US are combined, sonoporation is induced, which is an intracellular generation of ROS, or the energy stimulation of aquaporins to deliver ROS.

In addition, the injection of extracellular ROS into sonoporated cells has been identified as a primary method in [69]. The growing demand for safe, fresh, and healthy food necessitates the exploration of innovative non-thermal processing methods. PAW, in combination with other technologies like US, has emerged as a promising tool for inactivating food-borne pathogens and extending shelf life while preserving product quality. Further research is crucial to optimize the parameters of these combined approaches and ensure their effective implementation in the food industry.

## 5. Conclusions

This study investigated the optimal conditions for inactivating *E. coli* on chicken meat using a combined approach of PAW, MNB, UV, and US. The initial microbial disinfection trials revealed that PAW alone showed moderately impressive results against *E.*

*coli*. However, for chicken meat applications, the combination of PAW, MNB, UV, and US emerged as the most effective strategy. PAW with MNB for 10 min followed by 20 min of UV achieved a 6.00 $\log_{10}$ CFU/mL reduction in *E. coli* on chicken meat under the optimal soaking condition. PAW, UV, and US were all identified as crucial factors ($p < 0.05$) impacting bacterial survival, thus highlighting the synergistic effectiveness of these techniques. Combining PAW with other technologies not only proved effective in enhancing bacteria inactivation on chicken meat compared to PAW alone, but also suggests the potential for further improvements in food processing.

**Author Contributions:** Conceptualization, K.M., P.S., D.B. and W.W.; Methodology, K.M., P.S. and W.W.; Investigation, K.M., P.S. and W.W.; Data curation, W.W.; Writing—original draft, K.M.; Writing—review & editing, K.M., P.S. and W.W.; Supervision, P.S., D.B. and W.W.; Funding acquisition, W.W. All authors have read and agreed to the published version of the manuscript.

**Funding:** This research work was partially supported by Chiang Mai University.

**Data Availability Statement:** Data are contained within the article.

**Conflicts of Interest:** The authors declare no conflict of interest.

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
