# Peer review of "Synergistic Effect of Plasma-Activated Water with Micro/Nanobubbles, Ultraviolet Photolysis, and Ultrasonication on Enhanced Escherichia coli Inactivation in Chicken Meat"

_processes, doi:10.3390/pr12030567_

Round 1
Reviewer 1 Report
Comments and Suggestions for Authors
The reviewed manuscript presented interesting results of using plasma-activated water in combination with other techniques for inactivating some high-risk pathogens in fresh chicken meat. I think the manuscript has scientific value but needs a clearer presentation and highlighting of the practical application of the proposed approach. I have some specific suggestions for improvement of manuscript quality and presentation:
1) The introduction needs improvement - especially in the language used. Text in places was not clear to me and I was confused by some sentences (pay attention to lines 37-39; 62-63). The paragraph (49-54) must be revised - it is better to write the bacterial names whole.
2) The Materials and Methods section is well presented but I recommend the presentation of Fig. 1 to be improved by including a schematic diagram.
3) Some of the figures and tables need to be revised (Fig. 4, 5, Table 3) Authors should consider that figures and tables are standalone items. All the abbreviations/terms used should be described in captions or legends.
4) The discussion needs to be more targeted. It is unnecessarily long but still does not provide a complete discussion of the results and more importantly - a comparison with other similar studies.
Comments on the Quality of English LanguageLanguage needs to be revised. There are several confusion in word use and other grammatical errors.
Author Response
Dear Reviewer,
I trust this email finds you well. I am writing to update you on the changes made to the manuscript entitled “Integration Plasma-activated water with Supplementary Technique for Escherichia coli Inactivation in Chicken Meat" following the valuable feedback provided by the reviewer.
In response to the reviewer's suggestions, we have made significant revisions to the manuscript. These modifications were aimed at addressing the concerns raised and enhancing the overall quality of the content. We believe that the adjustments made will contribute to the improvement of the manuscript and align it more closely with the journal's scope and objectives.
Below, I have outlined the key changes that have been implemented:
Title Change:
Previous Title: "Integration Plasma-activated water with Supplementary Technique for Escherichia coli Inactivation in Chicken Meat"
Revised Title: " Synergistic Effect of Plasma-Activated Water with Micro/Nanobubbles, Ultraviolet Photolysis, and Ultrasonication for Enhanced Escherichia coli Inactivation in Chicken Meat "
The manuscript has been proofread by “ProofreadingServices.com” (certificate attached)
A thorough review of the content has been conducted, with particular attention to the points highlighted by the reviewer.
We sincerely appreciate the constructive feedback provided by the reviewer and the editorial team. We believe that the revised manuscript now aligns more closely with the objectives of the “Processes” journal, and we are confident that these changes will enhance the overall contribution of the paper.
Attached to this email, please find the revised manuscript and the responses to reviewers for your review. We hope that you find the modifications satisfactory, and we look forward to any further guidance or suggestions you may have.
Thank you for your time and consideration.
Best regards,
Wassanai

Reviewer 2 Report
Comments and Suggestions for Authors
The manuscript proposed a method to increase the efficiency of sterilization by using Plasma-activated water (PAW) together with supplementary technologies, including Micro/Nano-bubbles (MNB), Ultraviolet (UV) Photolysis, and Ultrasonication (US). The results revealed that integrating PAW with appropriate supplementary technologies increased inactivation efficiency by 97% and could enhance disinfection efficiency in chicken meat, providing an alternative method for pathogen inactivation in the poultry industry. The research has shown interesting and important finding. However, there are several questions need to be answered before I would recommend publish it. Details of my comments are following:
1. This sentence in the Introduction needs further explanation “However, chicken meat is thick, complex, and contains fat during processing, which compromises its sterilization efficiency in the pountry industry.”. The authors pointed out that chicken meat due to its own conditions lead to PAW on its sterilization efficiency is relatively low, so what is the sterilization rate of plasma used for meat sterilization, which is of great significance for the application of plasma co-processing technology to the meat industry in this paper.
2. There are no space between numbers and units in the manuscript, which are inappropriate. And please check the whole text again and pay attention to the units.
3. The horizontal lines in Tables 3 and 6 are not coordinated, please improve it.
4. Please change the E. coli to italics. And please change the p value to P.
5. Do not repeat the definition of a noun abbreviation, only the first time it occurs, such as ROS.
Comments on the Quality of English LanguageThe English shoud be improved.
Author Response

(The authors gave the same response as above.)

Reviewer 3 Report
Comments and Suggestions for Authors
This study proposed a method to increase the efficiency of sterilization by using Plasma-activated water (PAW) together with supplementary technologies, including Micro/Nano-bubbles (MNB), Ultraviolet (UV) Photolysis, and Ultrasonication (US). It represents some significances, however, there are some issues need to be carefully thought.
1. The word "water" in the title should be capitalized;
2. The reason for why combining the 4 technologies to do the treatment need to be further elucidated, researhces have showed that using PAW itself can already exert a great disinfection effect;
3. The results are not sufficent to present the mechanism behind the reduction, why did this effect happen need to be further explained.
4. Only the three or four technologies combination show a great log reduction, will this be meaningful if we use these expensive technologies all together, need to be clarified.
5. The way presenting graph need to be improved.
Comments on the Quality of English Language
Minor editing of English language required
Author Response

(The authors gave the same response as above.)

Round 2
Reviewer 1 Report
Comments and Suggestions for Authors
Dear Authors,
Dear Editors,
The quality of the manuscript has improved significantly. I recommend publication.
Best regards
Comments on the Quality of English LanguageMy only recommendation at this stage is another editorial reading of the English usage - please note lines 236-237.
Author Response
|
3. Point-by-point response to Comments and Suggestions for Authors |
|
Comments 1: The quality of the manuscript has improved significantly. I recommend publication.
Comments on the Quality of English Language My only recommendation at this stage is another editorial reading of the English usage - please note lines 236-237.
|
|
Response 1: Lines 236-239: The surviving population of E. coli after different plasma treatment conditions. According to the results, the experiment of PAW/MNB/UV and PAW/MNB/UV/US could reduce E. coli the most by 6.00 log10 CFU/ml, as shown in Figure 4d.] Thank you for pointing this out. We agree with this comment. Therefore, we have changed to “The combination techniques in the treatment of PAW/MNB/UV and PAW/MNB/UV/US proved highly effective, resulting in a 6.00 log10 CFU/ml reduction in E. coli, as shown in Figure 4d.”
|
Reviewer 3 Report
Comments and Suggestions for Authors
I am fine with current version.
Author Response
|
3. Point-by-point response to Comments and Suggestions for Authors |
|
Comments 1: I am fine with current version.
|
|
Response 1: Thank you very much for taking the time to review this manuscript.
|